An analysis of structural relationship among achievement motive on social participation, purpose in life, and role expectations among community dwelling elderly attending day services

Sano Nobuyuki 1 sanokichi09094@gmail.com
Kyougoku Makoto 2
1 Graduate School of Health Sciences, Kibi International University , Takahashi, Okayama , Japan
2 Department of Occupational Therapy, School of Health Sciences and Social Welfare, Kibi International University , Takahashi, Okayama , Japan
Ploughman Michelle
Electronic publication date: 2016 Jan 28
Publication date: 2016
Volume: 4
Electronic Location ID: e1655
Received 2015 Jul 6; Accepted 2016 Jan 13
Copyright: ©2016 Sano and Kyougoku
Copyright year: 2016
Copyright holder: Sano and Kyougoku
License: This is an open access article distributed under the terms of the Creative Commons Attribution License, which permits unrestricted use, distribution, reproduction and adaptation in any medium and for any purpose provided that it is properly attributed. For attribution, the original author(s), title, publication source (PeerJ) and either DOI or URL of the article must be cited.
License URL: https://creativecommons.org/licenses/by/4.0/

Keywords: Achievement motive, Social participation, Ikigai, Role expectation, Community-dwelling elderly people, Rehabilitation, SAMR, Structural equation modeling approach

Funding: The authors received no funding for this work.

==============================
Background. Achievement motive is defined as the intention to achieve one’s goals. Achievement motive is assumed to promote clients to choices and actions toward their valuable goal, so it is an important consideration in rehabilitation.

Purpose. The purpose of this study is to demonstrate the structural relationship among achievement motive on purpose in life, social participation, and role expectation of community-dwelling elderly people.

Methods. Participants were community-dwelling elderly people in day-service centers. A total of 281 participants (male: 127, female: 154) answered the self-administered questionnaire in cross-sectional research. The questionnaire was comprised of demographic data and scales that evaluated achievement motive, social participation, purpose in life, and role expectation. We studied the structural relationship established by our hypothesized model via a structural equation modeling approach.

Results. We checked the standardized path coefficients and the modification indices; the modified model’s statistics were a good fit: CFI = 0.984, TLI = 0.983, RMSEA = 0.050, 90% CI [0.044–0.055]. Achievement motive had a significantly direct effect on purpose in life (direct effect = 0.445, p value < 0.001), a significantly indirect effect on purpose in life via social participation or role expectation (indirect effect = 0.170, p value < 0.001) and a total effect on purpose in life (total effect = 0.615).

Discussion. This result suggests that enhancing the intention to achieve one’s goals enables participants to feel a spirit of challenge with a purpose and a sense of fulfillment in their daily lives.

Introduction

Achievement motive was proposed as a concept of psychogenic needs by Murray (1938) and, was defined as “a recurrent need to improve one’s past performance” by McClelland (1987). Achievement motive has been associated with various concepts such as intrinsic motivation, achievement goal, and self-efficacy (Bandura, 1977; Nicholls, 1984; Ryan & Deci, 2000). These concepts share the idea that individual motives are required to achieve goals although they differ with respect to theoretical background and the phenomena that researchers try to understand; for example, intrinsic motivation assumes that all people are inclined to participate in independent and active learning (Ryan & Deci, 2000), and self-efficacy additionally involves the implicit assumption that all people have a desire to be competent (Bandura, 1977). Thus, these terms have a common meaning with regard to achieving the same goal.

In rehabilitation, achievement motive is a significant concept concerning the assessment and intervention of clients (Lampton, Lambert & Yost, 1993; Resnick et al., 2002; Spivack et al., 1982; Vanetzian, 1997). The World Health Organization (WHO) defined the rehabilitation of people with disabilities as “a process aimed at enabling them to reach and maintain their optimal physical, sensory, intellectual, psychological, and social functional levels. Rehabilitation provides disabled people with the tools they need to attain independence and self-determination” (WHO, 2015). To enable them to attain independence and self-determination, it is important to assesse and treat the unique experiences, goals, and values of each person in the person-centered therapy (Cloninger & Cloninger, 2011). In other words, rehabilitation assumes to motivate and coach clients to be able to make self-directed choices for their own well-being and take initiative for achieving their own autonomy. Therefore, achievement motive focuses on the strength of the needs to improve based on experiences, goals, and values of clients.

Recently, the International Classification of Functioning, Disability and Health (ICF) emphasized that rehabilitation with an “Activity” and “Participation” framework is important, especially for elderly people, because the conventional rehabilitation for elderly people has been insufficient in improving the general functioning involved with activity and participation (Tsuruta, 2015). Activity and participation is the valuable; however, they should be task- and action-oriented that is executed originally or attempted to be execute anew. In support of the concept of activity and participation, it is necessary to enable elderly people to live with a purpose in life (“Ikigai” in Japanese) and fulfill role expectations in their lives (Tsuruta, 2015). Purpose in life refers to the tendency to derive meaning from life’s experiences and possess a sense of intentionality and goal directedness that guides behavior (Boyle et al., 2009; Boyle, Buchman & Bennett, 2010). Previous studies of purpose in life have been shown to be associated with psychological outcomes such as well-being and happiness (Ryff et al., 2006; Ryff & Keyes, 1995; Samman, 2007). Also, the study for middle-aged and elderly people revealed that purpose in life positively affected social participation over a 6-month period (Imai, 2013). Social participation is defined in terms of the consequences of activities in the social environment (Bukov, Maas & Lampert, 2002). Furthermore, purpose in life involves role expectation, which is defined as an understanding of one’s contribution to society and others (Demura & Sato, 2006). Moreover, the study on community-dwelling elderly people showed that achievement motive affected social participation and health-related quality of life (HRQOL, comprising of vitality, role physical, role emotional, and general health perception) positively (Sano, Kyougoku & Teraoka, 2015). Achievement motive is assumed to promote clients’ choices and actions toward their valuable goal. However, the effect of achievement motive on purpose in life, role expectation, and social participation is unclear.

Thus, we hypothesized that a high state of achievement motive leads to satisfaction in social participation, purpose in life, and role expectation for elderly persons. In addition, we hypothesized that social participation and role expectation promoted by achievement motive have an enhancing effect on purpose in life. Therefore, we created a hypothesized model demonstrating that achievement motive has a positive effect on purpose in life, role expectation, and social participation and that social participation and role expectation have a positive effect on purpose in life (Fig. 1).

Figure 1 Hypothesized model.

The purpose of this study was to demonstrate the structural relationship among achievement motive on purpose in life, social participation, and role expectation among community-dwelling elderly people. The structural relationship is a snapshot of a point in time and the observed findings suggest an influence on the variables (Teraoka & Kyougoku, 2015).

Methods

Ethics statement

This study employed a cross-sectional design. It was conducted in accordance with the Declaration of Helsinki and was approved by the Ethics Committee of the Kibi International University (No. 13-34). In addition, we gained approval from the facility directors of the institutions that cooperated in this study. We explained to participants that they could freely decide whether to participate in the study and could refuse to answer the questionnaire during this study. We completely protected the privacy of personal information. Furthermore, we obtained written informed consent from all participants. Participants put questionnaires in a box or handed them to staff.

Participants

Participants were community-dwelling elderly people in day-service centers. As the exclusion criteria, we excluded people who had been diagnosed with mental disorders such as schizophrenia and dementia, those who demonstrated clinical decline of cognitive function, and those who could not read or write the questionnaire forms.

Questionnaires

Demographic information

Demographic data such as gender, age, the name of the primary illness or disease, nursing care level (needing care: 1–5, needing support: 1–2 or nothing), the number of housemates, activities outside the home and hobbies, family structure, and subjective economic condition were obtained. The respective number of outside-the-home activities and hobbies were determined as follows: “How many times a week do you usually go outside your home?” and “How many hobbies do you have that continue to give you pleasure?” The subjective economic condition ranged from 1 to 4 (1 = I am economically stable and I don’t have to worry and 4 = I am poor and very nervous about my financial future) (Mizota et al., 2009).

SAMR (Sano & Kyougoku, 2015; Sano, Kyougoku & Yabuwaki, 2014)

We selected SAMR comprising 10 items to evaluate the state of achievement motive of clients and assumed in oblique 2-factor models: (a) Self-mastery–derived (six items), (b) Means/process-oriented–derived (four items). Each item in SAMR had a 7-point Likert scale ranging from 1 (strongly disagree) to 7 (strongly agree). The following is an example of an item: “I think that I can overcome any difficulty to achieve my goal.” (Appendix S1). The standardization score calculated depending on a total score, and if the achievement motive is strong then the standardization score will be high.

Self-completed occupational index (SOPI) (Imai & Saito, 2010; Imai & Saito, 2011)

We selected SOPI comprising nine items to evaluate the state of social participation of clients and assumed in oblique 3-factor models: (a) Leisure, (b) Productivity, (c) Self-care (each with three items). Each item in SOPI had a 5-point Likert scale ranging from 1 (I hardly have been satisfied) to 5 (I have been very satisfied). Summary score was calculated the following equation; (total score of 9 items −9)/36 × 100. The following in an example of an item: “Have you been able to perform important leisure activities in the past month?” SOPI was accepted for validity (concurrent) and reliability (internal consistency). If the quality of social participation is high then the summary score of SOPI will also be high.

K-1 scale for the feeling that life is worth living among the aged (K-1 scale) (Kondo, 2007)

We selected K-1 Scale comprising 16 items to evaluate the state of purpose in life and assumed in oblique 4-factor models: (a) Self-realization and will (six items), (b) Sense of life fulfillment (five items), (c) Will to live (two items), (d) Sense of existence (three items). Each item in K-1 Scale had a 3-point Likert scale ranging from 2 (yes) to 0 (no). The following is an example of an item: “I feel something to realize my accomplishment.” We reverse-scored item 2, 4, 9, and 12, phrased such that an agreement with the item represented a low level of purpose in life. K-1 Scale was accepted for validity (concurrent, factorial) and reliability (test–retest, internal consistency). If the quality of purpose in life is high, the total score of K-1 Scale will also be high.

Role expectation

We evaluated role expectation in a multiple-choice form. We provided 11 items for reference to a role checklist: volunteer, caregiver, housework, friend, family member, religionist, hobbyist or amateur, participant in an organization, student, worker, and other (Kielhofner, 2007). Participants selected the roles that were applicable to them. In the analysis, we counted the total number of chosen roles and aggregated choices of each role.

Statistical methods

Descriptive statistics and test of normality were conducted using SPSS Statistics 22 (http://www-01.ibm.com/software/jp/analytics/spss/products/statistics/). Item validity was conducted using Exametrika Version 5.3 (http://antlers.rd.dnc.ac.jp/ shojima/exmk/index.htm). Correlation between SAMR, SOPI, K-1 Scale and role expectation were conducted using HAD12 (http://norimune.net/had). Tests of structural validity and structural relationship were conducted using Mplus v7.2 (http://www.statmodel.com).

Descriptive statistics and test of normality

We performed simple descriptive statistics including means and standard deviation (SD) and calculated the Kolmogorov–Smirnov test, skewness, and kurtosis.

Items validity

We calculated the mean information content (entropy) and the total polyserial correlation coefficient (PCC) for all items of SAMR, SOPI, and K-1 Scale. A PCC value of >0.2 was the standard item validity (Toyoda, 2002).

Structural validity

We analyzed the factor structure of SAMR, SOPI, and K-1 Scale using confirmatory factor analysis (CFA) by a structural equation modeling (SEM) approach (Muthén, 1983) for the participation of this study. The factor structure of each scale was examined using the same factor structure as previous studies. We used the Maximum Likelihood with Robust standard error (MLR) with missing data for SAMR and SOPI and the modified weighted least squares method (WLSMV) with missing data for K-1 Scale. We referred to several fit indices: Comparative Fit Index (CFI), Tucker Lewis Index (TLI), and Root Mean Square Error of Approximation (RMSEA) with 90% confidence interval (CI). A CFI and TLI value of >0.9 was the best model fit. For RMSEA, values ≤0.05 indicate a close fit, those of ≤0.08 indicate a reasonable fit, and those of ≥0.1 indicate a poor fit (MacCallum, Browne & Sugawara, 1996).

Correlation between SAMR, SOPI, K-1 Scale, and role expectation

We calculated polychoric correlation, polyserial correlation or spearman correlation for subscale score, total scale score, and summary score of SAMR, SOPI, K-1 scale, and role expectation (total number of role item, each role item). Values of >0.2 and <0.4 indicate weak correlation, those of >0.4 and <0.7 indicate moderate correlation, and those of >0.7 and <0.9 indicate a strong correlation.

Structural relationship

We tested our hypothesized model (Fig. 1) using Multiple Indicator MultIple Cause (MIMIC) by a SEM approach. MIMIC is the model to verify a hypothesis that some observation variables affect several latent variables and the latent variables affect some different observation variables (Kosugi & Shimizu, 2014). This approach allowed us to evaluate how well our hypothesized relationships between a latent exogenous variable (achievement motive), latent mediators (social participation, role expectation), and a manifest endogenous variable (purpose in life) fit our data. We used the WLSMV with missing data for our analysis and referred to several fit indices: CFI, TLI, RMSEA, 90% CI. The standard of the best model fit was the same as that of structural validity. We also estimated the values of direct effect and indirect effect each with 90% CI.

Results

Participant characteristics

We recruited a total of 304 participants from 11 day-service centers that participated in this study. A total number of 281 participants answered the questionnaire (valid response rate: 92.4%); 127 (45.2%) were men and 154 (54.8%) were women, and mean age was 77.1 ± 8.7 years. Details of the sample characteristics are described in Table 1.

Table 1 Participant characteristics.

	Class	n = 281	%	
Gender	Male	127	45.2%	
	Female	154	54.8%	
Age (mean ± SD)	77.1 ± 8.7			
Disease	Orthopedic	111	39.5%	
	Neurological	108	38.4%	
	Heart	5	1.8%	
	Others	29	10.3%	
	Unknown	28	10.0%	
Care level	Care5	0	0.0%	
	Care4	8	2.8%	
	Care3	23	8.2%	
	Care2	74	26.3%	
	Care1	65	23.1%	
	Support2	59	21.0%	
	Support1	48	17.1%	
	Nothing	0	0.0%	
	Unknown	4	1.4%	
Housemate (mean ± SD)	1.6 ± 1.4			
Going out (mean ± SD)	4.0 ± 3.0			
Hobby (mean ± SD)	1.4 ± 1.3			
Spouse	With	160	56.9%	
	Without	121	43.1%	
Grandchildren	With	44	15.7%	
	Without	237	84.3%	
Economic condition	1	68	24.2%	
	2	172	61.2%	
	3	38	13.5%	
	4	2	0.7%	
	Unknown	1	0.4%	
Roles (mean ± SD)	1.5 ± 1.0			
	Volunteer	9	3.2%	
	Caregiver	3	1.1%	
	Housework	73	26.0%	
	Friend	46	16.4%	
	Family member	207	73.7%	
	Religionist	9	3.2%	
	Hobbyist	42	14.9%	
	Organization	15	5.3%	
	Student	0	0.0%	
	Worker	5	1.8%	
	Other	17	6.0%	
Notes.

Hobbyist Hobbyist or Amateur

Organization Participant in organization

Descriptive statistics and test of normality

Table 2 indicates descriptive statistics and normality tests of the three scales (SAMR, SOPI, and K-1). In a test of normality of each scales, total score of SAMR is 0.003 (skewness = −0.821, kurtosis = 1.560), summary score of SOPI is 0.069 (skewness = 0.100, kurtosis = −0.671), total score of K-1 Scale is 0.000 (skewness = −0.546, kurtosis = −0.514). Although the other variables had not an extreme deviation from the mean and SD, the items of SAMR and K-1 were needed attention in skewness and kurtosis.

Table 2 Descriptive statistics, and items validity.

Self-mastery-derived involves Item 1–6 of SAMR, Means/process-oriented-derived involves Item 7–10 of SAMR. Leisure involves Item 1–3 of SOPI, Productivity involves Item 4–6 of SOPI, Self-care involves Item 7–9 of SOPI. Self-realization and will involves Item1, 3, 5, 6, 14, 15 of K-1 Scale, Sense of life fulfillment involves Item2, 4, 8, 9, 12 of K-1 Scale, Will to live involves Item11, 13 of K-1 Scale, Sense of existence involves Item7, 10, 16 of K-1 Scale.

Item	Mean	SD	Entropy	PCC	
SAMR					
Item1	5.129	1.390	2.363	0.744	
Item2	5.089	1.332	2.290	0.697	
Item3	5.139	1.419	2.406	0.748	
Item4	4.723	1.442	2.468	0.825	
Item5	5.299	1.370	2.324	0.819	
Item6	4.750	1.389	2.407	0.748	
Item7	5.786	1.277	2.109	0.756	
Item8	5.505	1.300	2.248	0.694	
Item9	5.760	1.340	2.159	0.733	
Item10	4.707	1.637	2.592	0.571	
Self-mastery–derived	30.044	6.663			
Means/process-oriented–derived	21.754	4.323			
Total scale score	51.798	9.985			
SOPI					
Item1	2.950	1.183	2.220	0.865	
Item2	2.928	1.157	2.196	0.879	
Item3	2.871	1.219	2.232	0.883	
Item4	2.712	1.265	2.250	0.894	
Item5	2.688	1.238	2.237	0.910	
Item6	2.647	1.268	2.245	0.918	
Item7	3.208	1.217	2.210	0.818	
Item8	3.082	1.155	2.192	0.900	
Item9	3.072	1.233	2.257	0.860	
Leisure	8.763	3.360			
Productivity	8.054	3.670			
Self-care	9.362	3.461			
Summary score	47.782	25.666			
K-1 Scale					
Item1	1.354	0.821	1.408	0.587	
Item2	1.173	0.833	1.377	0.738	
Item3	1.421	0.764	1.275	0.689	
Item4	1.365	0.808	1.524	0.666	
Item5	1.482	0.773	1.556	0.683	
Item6	1.231	0.810	1.420	0.656	
Item7	1.159	0.819	1.542	0.713	
Item8	1.329	0.689	1.362	0.611	
Item9	1.397	0.786	1.281	0.707	
Item10	1.195	0.788	1.577	0.734	
Item11	1.441	0.701	1.542	0.744	
Item12	1.516	0.753	1.554	0.702	
Item13	1.504	0.733	1.544	0.588	
Item14	1.068	0.798	1.410	0.566	
Item15	1.187	0.824	1.392	0.607	
Item16	1.168	0.798	1.237	0.356	
Self-realization and will	7.785	3.396			
Sense of life fulfillment	6.797	2.612			
Will to live	2.950	1.237			
Sense of existence	3.538	1.983			
Total scale score	21.171	7.335			

Figure 2 CFA of SAMR.

CFI, 0.955; TLI, 0.941; RMSEA, 0.061; 90% CI [0.040–0.081]. Abbreviations of the four factors are similar to Table 3.

Items validity

All items for SAMR, SOPI, and K-1 Scale were accepted and the value satisfied the standard of PCC (Table 2).

Structural validity

CFA of SAMR, SOPI, and K-1 Scale demonstrated good fit statistics of the same structure with previous studies. Fit indices of SAMR were CFI = 0.955, TLI = 0.941, RMSEA = 0.061, 90% CI [0.040–0.081], and factorial correlation between Self-mastery–derived and Means/process-oriented–derived was 0.768 (Fig. 2). Fit indices of SOPI were CFI = 0.982, TLI = 0.976, RMSEA = 0.058, 90% CI [0.034–0.082], and factorial correlation between three factors was 0.731 (Leisure and Productivity), 0.598 (Leisure and Self-care) and 0.625 (Productivity and Self-care) (Fig. 3). Fit indices of K-1 Scale were CFI = 0.944, TLI = 0.932, RMSEA = 0.078, 90% CI [0.066–0.089] and factorial correlation between four factors was 0.670 (Self-realization and will and Sense of life fulfillment), 0.822 (Self-realization and will and Will to live), 0.813 (Self-realization and will and Sense of existence), 0.583 (Sense of life fulfillment and Will to live), 0.558 (Sense of life fulfillment and Sense of existence), and 0.804 (Will to live and Sense of existence) (Fig. 4). Although SAMR and K-1 Scale were not sufficient to test for normality, we comprehensively decided that all scales were available for examination of correlation between this study variable and structural relationship.

Figure 3 CFA of SOPI.

CFI, 0.982; TLI, 0.976; RMSEA, 0.058; 90% CI [0.034–0.082].

Figure 4 CFA of K-1 Scale.

CFI, 0.944; TLI, 0.932; RMSEA, 0.078; 90% CI [0.066–0.089]. Abbreviations of the four factors are similar to Table 3.

Correlation between SAMR, SOPI, K-1 Scale, and role expectation

We excluded the role of student and other because the number of students was 0 and descriptive contents of other were unspecified. Positive correlation was accepted between most of the subscale score, summary score, and total scale score of SAMR, SOPI, and K-1 scale (Table 3). Of the total number of role items, the roles of Friend, Hobbyist or Amateur, and Participant in an Organization were a positive correlation with SAMR, SOPI, and K-1 scale (Table 4).

Table 3 Correlation between SAMR, SOPI, and K-1 Scalele.

The values calculated by spearman correlation are on double line, the values calculated by polyserial correlation are on underline, and other values are calculated by polychoric correlation.

	Mastery	Means	SA Total	Leisure	Product	Self-care	Summary	
Leisure	0.388**	0.192 **	0.337 **					
Product	0.420 **	0.217 **	0.368 **					
Self-care	0.374 **	0.252 **	0.353 **					
Summary	0.419 **	0.237 **	0.377 **					
Realize	0.542 **	0.302 **	0.494 **	0.326**	0.291**	0.279**	0.323 **	
Fulfill	0.347 **	0.091	0.271 **	0.291**	0.253**	0.309**	0.308 **	
Will	0.401 **	0.253 **	0.379 **	0.286**	0.223**	0.161*	0.246 **	
Exist	0.404 **	0.288 **	0.407 **	0.281**	0.306**	0.310**	0.324 **	
K-1 Total	0.534 **	0.290 **	0.483 **	0.348 **	0.319 **	0.323 **	0.362 **	
Notes.

Mastery Self-mastery-derived

Means Means/process-oriented-derived

SA Total total scale score of SAMR

Product Productivity

Summary summary score of SOPI

Realize Self-realization and will

Fulfill Sense of life fulfillment

Will Will to live

Exist Sense of existence

K-1 Total Total scale score of K-1 scale

* p < .05.

** p < .01.

Table 4 Correlation between SAMR, SOPI, K-1 Scale, and role expectation.

Most of abbreviations are similar to Table 3. The values calculated by polyserial correlation are on underline, and other values are calculated by polychoric correlation.

	Mastery	Means	SA Total	Leisure	Product	Self-care	Summary	Realize	Fulfill	Will	Exist	K-1 Total	
Total roles	0.314 **	0.199 **	0.298 **	0.287 **	0.272 **	0.281 **	0.305 **	0.443 **	0.284 **	0.216 **	0.364 **	0.414 **	
Volunteer	0.217	0.015	0.150	0.141	0.064	0.019	0.090	0.325 *	0.371 *	–	0.198	0.463+	
Caregiver	0.178	0.320	0.256	0.097	0.050	−0.156	−0.008	0.345+	0.362+	0.161	0.226	0.324 +	
Housework	0.023	0.048	0.051	0.037	0.126	0.079	0.087	0.073	0.022	−0.100	0.179*	0.056	
Friend	0.313 **	0.154	0.274 **	0.320 **	0.308 **	0.372 **	0.372 **	0.452 **	0.231 *	0.049	0.257 **	0.405 **	
Family	0.018	−0.013	−0.004	0.050	0.021	−0.050	0.007	0.050	0.101	0.290 **	0.080	0.114	
Religionist	0.355 *	0.342	0.396 *	0.186	0.011	0.193	0.139	0.251+	0.160	0.186	0.244	0.271	
Hobbyist	0.390 **	0.311 **	0.399 **	0.377 **	0.295 **	0.399 **	0.387 **	0.458 **	0.303 **	0.131	0.257 **	0.422 **	
Organization	0.406 **	0.075	0.294 **	0.275 *	0.228+	0.220+	0.257 +	0.470 **	0.339 **	0.313 *	0.427 **	0.522 **	
Worker	0.079	0.291	0.176	0.033	0.194	0.089	0.111	0.446 *	0.403 *	0.353	0.528 **	0.534	
Notes.

Total roles Total number of roles

Family Family member

Hobbyist Hobbyist or Amateur

Organization Participant in organization

+ p < .10.

* p < .05.

** p < .01.

In particular, the subscale of Self-mastery–derived and total scale scores of SAMR was moderately correlated with the subscale score of Self-realization and will, Sense of existence, and total scale scores of K-1 Scale (0.404–0.542). The roles of volunteer, friend, hobbyist or amateur, participant in an organization, and worker were moderately correlated with the subscale score of K-1 Scale or a total scale score of K-1 Scale (0.403–0.528).

Figure 5 Hypothesized model using SEM.

CFI, 0.986; TLI, 0.985; RMSEA, 0.047; 90% CI [0.042–0.053]. Most of abbreviations are similar to Table 3. Role Expectation, total number of roles; SA, items of SAMR; SO, items of SOPI; KS, items of K-1 Scale. The error terms are omitted to make the figure simple.

Structural relationship

The hypothesized model using SEM was good fit statistics: CFI = 0.986, TLI = 0.985, RMSEA = 0.047, 90% CI [0.042–0.053] (Fig. 5). However, the standardized path coefficient that achievement motive configured as the dominant conception of two factors of SAMR affecting Self-mastery-derived was beyond 1.0 (1.099), and the hypothesized model was suspected to involve model misspecification. The correlation of two factors of SAMR was very strong; therefore, a problem of linear dependence between these two factors or these items may occur, similar to that in previous studies (Sano, 2014). To revise the model by equality constraints associated with Self-mastery-derived, we restricted the standardized path coefficients of Self-mastery–derived on the factor’s items and achievement motive on two factors of SAMR were to 1. As a result, the modified model was good fit statistics: CFI = 0.984, TLI = 0.983, RMSEA = 0.050, 90% CI [0.044–0.055] and did not involve standardized path coefficients beyond 1.0 (Fig. 6). With respect to the standardized path coefficients in the modified model, achievement motive (direct effect = 0.445, p value = 0.000), social participation (direct effect = 0.161, p value = 0.015), and role expectation (direct effect = 0.224, p value = 0.000) had a significant positive impact on purpose in life; achievement motive (direct effect = 0.499, p value = 0.000) had a significant positive impact on social participation; and achievement motive (direct effect = 0.400, p value = 0.000) had a significant positive impact on role expectation. Achievement motive (indirect effect = 0.080, p value = 0.018, 95% CI [0.014–0.147]) had a significant positive effect on purpose in life via social participation, and achievement motive (indirect effect = 0.089, p value = 0.000, 95% CI [0.043–0.136]) had a significant positive effect on purpose in life via role expectation. The sum of indirect effect was standardized path coefficients = 0.170, p value = 0.000, 95% CI [0.079–0.260]. The total effect of the achievement motive on purpose in life was standardized path coefficients = 0.615 (direct effect = 0.445 + indirect effect = 0.170).

Figure 6 Modified model using SEM.

CFI, 0.984; TLI, 0.983; RMSEA, 0.050; 90% CI [0.044–0.055]. Abbreviations are similar to Table 3 and Fig. 5. Standardized path coefficients of Self-mastery—derived on the factor’s items and achievement motive on two factors of SAMR were restricted to 1. The error terms are omitted to make the figure simple.

Discussion

The purpose of this study was to demonstrate the structural relationship among achievement motive on purpose in life, social participation, and role expectation of community-dwelling elderly people. We were able to show statistical evidence according to our hypothesis. Moreover, item validity and structural validity of SAMR, SOPI, and K-1 Scale were acceptable in this study. That is, these assessments function well as tools for quantification.

In the test of a structural relationship based on our hypotheses, it was demonstrated that achievement motive had a positive impact on purpose in life, social participation, and role expectation. Moreover, social participation and role expectation had a positive impact on purpose in life. We proved the strong effect of achievement motive on outcome indices of elderly persons. In addition, we confirmed the significantly indirect effects of achievement motive on purpose in life via social participation or role expectation. We could expect that purpose in life became enhanced through improvements of social participation or role expectation by achievement motive. These results suggest that enhancing the intention to achieve one’s goals allows participants to feel a spirit of challenge with a purpose and a sense of fulfillment in daily living. At the same time, recognizing engagement in important activities for oneself and the role of oneself in society also helps participants feel capable of being helpful to others.

The direct effect of social participation and role expectation on purpose in life was not significantly high. The result indicated that achievement motive had greater effects on the support-related purpose in life for elderly persons than on social participation and role expectation. In previous studies, purpose in life has been associated with well-being and life satisfaction (Bronk et al., 2009; Ryff, 1989; Ryff & Keyes, 1995), and has had an effect on restraining mortality risk, anxiety, and stress (Boyle et al., 2009; Ishida, 2012). Also, purpose in life is future-orientated and goal-orientated (Bronk et al., 2009), and involves the individual’s intention to achieve something (Demura & Sato, 2006; Nomura, 2005); therefore, enhancing purpose in life may be the goal for the rehabilitation of elderly persons. Thus, we considered the importance of pursuing the achievement of individual goals and ways of life. Establishing and sharing goals between client and therapist based on consultation about ideals and meaning in life is very important in rehabilitation.

Moreover, achievement motive has a positive correlation with purpose in life, social participation, and role expectation because these significant correlations were accepted among SAMR, SOPI, K-1 Scale, and the total number of role items. In particular, Self-mastery–derived was closely related to purpose in life due to moderate correlation with the subscale score and total scale score of K-1 Scale. Accordingly, we suggest that it is important to support clients in rehabilitation by enhancing their own abilities and intelligence through training, feedback, etc.

Regarding the correlation between SAMR, SOPI, K-1 Scale, and each of the roles, achievement motive, social participation, and purpose in life were not associated with the role in the home (Caregiver, Housework, and Family member). On the other hand, these concepts were positively associated with most roles related to society (friend, hobbyist or amateur, participant in an organization, and worker). Therefore, we suspect that roles within the home do not have much of an effect on the health care of community-dwelling elderly people. In contrast, we expect that it is more effective to support community-dwelling elderly people in roles related to their relationship with society.

Contribution and Limitation

This study proved the structural relationship among achievement motive on purpose in life, social participation, and role expectation of community-dwelling elderly people. We believe that this study reveals the far-reaching effects of achievement motive. Although achievement motive has not been sufficiently studied, it is considered an important element in rehabilitation (Resnick, 1996). Therapists who perform rehabilitation may be able to share and collaborate with others in attaining the goal of helping clients from this new standpoint of achievement motive.

This study has a few limitations on the study design. First, this study utilized data sampling and research for the participants restricted to day-service centers in specified areas. Second, this study used a self-reported questionnaire to collect data and could examine only subjective effects but could not examine effects by objective data indices. We hope to continue this study while considering these limitations. For example, elucidating the structural relationship between genders and among ages would be useful in future studies.

Supplemental Information

Supplemental Information 1 Raw data

Click here for additional data file.

Appendix S1 Scale for Achievement Motive in Rehabilitation (SAMR)

Click here for additional data file.

We wish to thank the elderly people who participated in this study, the staff in the day-service centers who cooperated, colleagues in the laboratory who advised us, and the families who supported us.

Additional Information and Declarations

Competing Interests

Author Contributions

Human Ethics

Data Availability

The authors declare there are no competing interests.

Nobuyuki Sano conceived and designed the experiments, performed the experiments, analyzed the data, contributed reagents/materials/analysis tools, wrote the paper, prepared figures and/or tables.

Makoto Kyougoku conceived and designed the experiments, analyzed the data, contributed reagents/materials/analysis tools, wrote the paper, reviewed drafts of the paper.

The following information was supplied relating to ethical approvals (i.e., approving body and any reference numbers):

This study was conducted in accordance with the Declaration of Helsinki and was approved by the Ethics Committee of the Kibi International University (No. 13-34). In addition, we gained approval by the facility directors of the institutions that cooperated in this study. We explained to participants that they could freely decide whether to participate in the study and could refuse to answer the questionnaire during this study. We completely protected the privacy of personal information. Furthermore, we obtained written informed consent from all participants.

The following information was supplied regarding data availability:

The research in this article did not generate any raw data.

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
