# Peer review of "An analysis of structural relationship among achievement motive on social participation, purpose in life, and role expectations among community dwelling elderly attending day services"

_PeerJ, doi:10.7717/peerj.1655_

## Round 0.1 · original submission · Major Revisions

· Academic Editor

Major Revisions

In addition to the points made by the reviewers,
Title: The title is not clear and somewhat misleading. The sample is of community-dwelling elderly people attending day services. There is no need to state "by using cross-sectional research" in the title. I would recommend "The effect of achievement motive on social participation, ikigai and role expectations among community dwelling elderly attending day services".

Abstract: Please provide complete p values, for example p<0.001 or p<0.0001 and so on
Introduction: The authors essentially review their own work without placing their current research in context of the broader evidence in this field. For example 'achievement motive' seems to align with self-efficacy which has been extensively studied in rehabilitation. Ikigai, as a concept, has parallels with self-reliance, self-efficacy and resilience: all extensively studied. It is not clear what the authors research adds to the body of literature. There is excessive descriptions of statistical data in the introduction section.

Please correct grammitcal errors in the Acknowledgments section.

·

Basic reporting

First of all let me thank you for the opportunity to read this interesting article.

I think the manuscript is publishable. However, I recommend a major revision of the article. At this moment is difficult to understand the theoretical basis of the achievement motive model the authors want to test. It seems that the article focus on the factor structure of a measure for this model but the study is not presented in that way. Rather the authors suggest a relationship to social participation, ikigai (which by the way needs to be introduced as a concept, this was, at least for me, a new notion), and role expectations.

At the same time, the authors present a factor structural validation already in the introduction, which confused me. Is this the same sample used in this article? The one the authors detail in the introduction comes from a conference presentation, which is fine, but please clarify if this is the same sample or not...in what way do they differ and why it is necessary to validate the structure of the measure in their present study.

In short, the authors need to re-organize their introduction by giving the theoretical background to their model and the expected associations to social participation, ikigai, and role expectations. After defining this concepts and their expected relationship, the authors may develop/introduce the scale they created and also motivate its creation. Then the authors will be in a position to clearly state the aims of their study: factor structure validation of the scale and associations to dependent variables.

Please observe that I'm not sure if the authors actually really aim to test the structure of the scale...if they do, then motivate and explicitly develop this as an aim. Otherwise, the factor structure validation might as well be a supplementary material to the article.

Experimental design

This is fine. However, the authors need to revise the whole paper to not overstate what they actually can study. For instance, in the abstract they state that "...previous studies have not demonstrated a causal relationship between achievement motive and a more enhanced..." This study is cross-sectional, so none of the conclusions can be causal. Please revise the whole paper for these type of statements. They do this partially in the limitation section but I suggest that these type of statements need to be avoided across the whole paper.

Validity of the findings

Please see my comments regarding "causality" statements in the present version.

Reviewer 2 ·

Basic reporting

1. Adherence and English
The manuscript adheres to the policies, and English is clear and might be conformed to the professional standards with minor change of expressions.

2. Abstract
In general, abstract needs not author’s detail opinion. From line 30 to 34 could be omitted in the abstract.

3. Introduction
I hope the authors make more effort to revise the introduction, because it was mainly introducing about the “Achievement motive” and its scale developed by the authors (I think it is better omitting line 71-82, which include statistics results in their previous study). However, it was lack of showing how work fits into the broader field of knowledge. It is useful to add evidence which links achievement motive (or similar indicators for which readers can understand) with health, especially for elderly person.
At first mention, the term ikigai (foreign words are normally italicized) should be translated in other ways, including“a sense of purpose in life”or “meaningful of life”. Because ikigai in Japanese is hard to understand clearly in foreign languages.

4. Relevant prior literature in references
They used a lot relevant prior literatures, but there are more than two times literatures written in Japanese comparing those in English. Consideration for the international journal’s readers, they had better using English one as possible. For example, prior study for ikigai or sense of meaning, some studies have been reported in English.

5. The structure of the submitted article
The volume of structure in the manuscript is a bit unbalanced. Introduction has 63 lines, Methods: 127, Results: 71, and Discussion: 44. In general, the discussion is important for readers to understand wider context for implicating their field. I recommend the Methods should be written simpler, and reduce the volume. Instead, the discussion needs revision in order to discuss more about the results of direct effect from the achievement motive on the ikigai, which is Japanese original emotional health, comparing with worldwide prior studies.

6. Tables and Figures
As a whole, they are relevant to the content of the article.
・In table 2, consider more explicitly relevant meaning of the Item, using keywords explaining each Item are useful for readers.
・Table 2 of descriptive statistics and test of normality is important but not essential for showing at a table. Instead, as a community-dwelling elderly people’s health study, showing distribution of the participants answer for the questionnaire might be useful for the readers.
・In figure5, 6 of the SEM models
1. Please add R2 for each endogenous latent variables.
2. Make all the standardized path coefficients are shown by two-digit number (ex. 0.2 →0.20).
3. Please add in the note that error terms are omitted in the models.

7. Results
Statistical way was conducted as appropriately in SEM analysis. However, modified model was lack of explaining its validity. Readers are not necessarily good at its analysis.

Experimental design

The manuscript could be considered as a study in Health Sciences. Research question, to clarify the structural relationship among the achievement motive was meaningful for the elderly people’s health. Their research have been conducted in conformity with the standards in the public health field.

Validity of the findings

The data was robust and analyzed standard way. The final model (Fig.6) for this manuscript’s conclusions was connected to the original question. However, the model needs more explanation to get acceptable. Because, they have modified the model (all paths from “Mastery” to its observed variables are 0.78) lacking for explaining their validity. If the authors revise to add explanation about the modification more clearly, Fig.5 might be omitted to prevent reader’s confusion.

Additional comments

The authors made good effort to analyze the structural relationship in detail and report precisely especially for the statistics results. However, as they mentioned in the limitation, the present study was “difficult to confirm causal relationships”. So I think they should use term “structural relationships” in the manuscript and title, instead of “causal”. SEM analysis permits to express structural relationship if data has no time passage.
Elucidating structural or causal relationship between men and women or among age groups would be needed in future study. Adding 1-2 sentences on this might be useful.

---

## Round 0.2 · Major Revisions

· Academic Editor

Major Revisions

The findings and impact of the study are important however the main issue now is with grammar and style. The writing is awkward, difficult to understand and distracting. Since almost every paragraph in the Introduction and Discussion needs editing, the authors should receive assistance from a professional editor before resubmitting.

For example:

Line 60: “Therefore, therapists expect that a higher achievement motive indicate more effective support for the client.” It is not clear what is meant here. Please state more precisely what you mean.
Line 66 “the treated phenomena”. This phrase does not make sense. Please revise.
Line 78 “SAMR can perceive achievement motive…” The tool discussed does not ‘perceive’ however it can measure or detect.
Line 233 “We examined the structural validity in the participation of this study..”
Line 400 “the significant correlations of achievement motive, social participation, and purpose in life were almost unrecognized with the role in the home” “On the other hand, the significant positive correlation of these concepts was recognized with roles related to society: The use of the word ‘recognized’ is not appropriate in these sentences.
Figures 2,3 and 4 are not necessary. It is sufficient to describe the distribution briefly in the Results.

·

Basic reporting

Thank you for the opportunity to review this second revised version of this manuscript. I think that the authors have moved forward. I really appreciate that the authors took away the methodological development of their scale from the Introduction. However, I still suggest a major revision before publication.

It is difficult to be specific. The authors have actually made a good work and the Introduction is less technical than it was in the last version. Nevertheless, the authors make something that is very common when writing academic papers; this includes myself, namely that they are very fast to talk about measures and variables already in the introduction, rather than talking about concepts and/or theoretical frameworks.

This has an effect on how clear the Introduction is because it lacks a logical development and understanding on why they do the study from the beginning using the concepts they use. For instance, the authors state (page 7, lines 82-83 in the pdf):

“Furthermore, rehabilitation is intended to enable the elderly people to live with a purpose in life (“Ikigai” in Japanese) and a fulfill role expectations in their lives.”

However, purpose in life is nowhere to be found in the definition of rehabilitation they give in the very beginning (page 5, lines 44-50 in the pdf):

“The World Health Organization (WHO) defined the rehabilitation of people with disabilities as “a process aimed at enabling them to reach and maintain their optimal physical, sensory, intellectual, psychological, and social functional levels. Rehabilitation provides disabled people with the tools they need to attain independence and self-determination” (WHO 2015). That is, rehabilitation assumes that the client takes the initiative for setting and achieving goals. Therefore, therapists expect that a higher achievement motive indicate more effective support for the client.”

If the authors mean to say that “..independence and self-determination” is similar or equal to purpose in life they need to make this explicit and at the very least develop the logic that lead them to this statement. In the very same sentence above the authors reword the WHO definition for rehabilitation as “rehabilitation assumes that the client takes the initiative for setting and achieving goals”. This is not, according to me, a correct rewording of the WHO definition for rehabilitation. For instance, in the definition by the WHO you find the following keywords/sentences: “enabling them” and “..provides disabled people with the tools they need…”. Here, I guess that the authors actually mean that rehabilitation aims to motivate/coach clients to be able to take initiative for achieving their own autonomy and make self-directed choices for their own well-being. This should indeed make their life more purposeful and meaningful (see for example the work by C. R. Cloninger). However, nothing of what I write here is explicit in the text. The authors might have another rationale behind the cause of their investigation.

In short, why is purpose in life investigated in relation to achieving motive? Stating that rehabilitation is intended to enable “Ikigai” is not only incongruent with the WHO definition the authors presented, the rationale is poorly developed for the investigation they conducted. A good example of what the authors should do is to work with the text as they did when introducing self-efficacy and intrinsic motivation. Revise the discussion also in light of the changes you make to the Introduction. All of the above are just a couple of examples. The Introduction needs to always adhere to why the authors are investigating EACH of the variables at hand in relation to achieving motive. The development of the rationale needs to be fluid across the Introduction.

Some minor suggestions follow next. The abstract should include a Conclusion section rather than a Discussion section. Please avoid acronyms to refer to concepts such as quality of life. Revise for language after any changes.

Experimental design

This is fine.

Validity of the findings

Good that the authors talk of relationships rather than causality.

Reviewer 2 ·

Basic reporting

No comments

Experimental design

No comments

Validity of the findings

No comments

Additional comments

The authors made good effort to revise their manuscript, and it could be acceptable. Then, I’ll show some recommendations as a final request for the authors.
Introduction has become clearly, and had a lot of prior literatures written in English. But they used a few kinds of explanations for “effect”, “affect”, “influence”, and “structural relationship”. Through the whole manuscript, including the title, I recommend to make more attention to standardize them as possible.
In results, the most important model figure might be Fig.9, not Fig.5, 6, 7, 8. So I think Fig.9 would not be printed as same size figure as the other, i.e. Fig.9 is a big size figure comparing Fig. 5-8.
Thank you for the authors’ good work, and thank for the editor making me review chance for the interesting manuscript.

---

## Round 0.3 · accepted · Accept

· Academic Editor

Accept

The authors have substantially improved this manuscript. There are still some minor grammatical issues however they may be able to be dealt with during the proof stage. eg the sentence on lines 68-69 does not make sense.